# Trash or Treasure? An Interactive Dual-Stream Strategy for Single Image Reflection Separation

**Qiming Hu, Xiaojie Guo**[*]

College of Intelligence and Computing, Tianjin University, Tianjin 300350, China

huqiming@tju.edu.cn xj.max.guo@gmail.com

## Abstract

Single image reflection separation (SIRS), as a representative blind source separation task, aims to recover two layers, *i.e.*, transmission and reflection, from one mixed observation, which is challenging due to the highly ill-posed nature. Existing deep learning based solutions typically restore the target layers individually, or with some concerns at the end of the output, barely taking into account the interaction across the two streams/branches. In order to utilize information more efficiently, this work presents a general yet simple interactive strategy, namely *your trash is my treasure* (YTMT), for constructing dual-stream decomposition networks. To be specific, we explicitly enforce the two streams to communicate with each other block-wisely. Inspired by the additive property between the two components, the interactive path can be easily built via transferring, instead of discarding, deactivated information by the ReLU rectifier from one stream to the other. Both ablation studies and experimental results on widely-used SIRS datasets are conducted to demonstrate the efficacy of YTMT, and reveal its superiority over other state-of-the-art alternatives. The implementation is quite simple and our code is publicly available at *https://github.com/mingcv/YTMT-Strategy*.

## 1 Introduction

Blind source separation, a long-standing problem in signal processing, aims to recover multiple intrinsic components from their mixture, the difficulty of which comes from its ill-posedness, *i.e.*, without extra information, there is an infinite number of feasible decompositions. Particularly in computer vision, image reflection separation (IRS) is a representative scenario that often occurs when taking pictures through a transparent medium such as glass. In such cases, the captured images will contain both the scene transmitted through the medium (transmission) and reflection. On the one hand, reflections are annoying for high-quality imaging, and may interfere with the performance of most, if not all, of classic and contemporary vision oriented algorithms such as object detection and segmentation. On the other hand, one may also want to see what happens in the reflection. Hence, developing effective transmission-reflection decomposition techniques is desired.

Formally, the captured superimposed image $I$ can be typically modeled as a linear combination of a transmission layer $T$ and a reflection $R$, *i.e.* $I = T + R$.[1] Over last decades, a large number of schemes have been devised to solve the decomposition problem. Various statistical priors and regularizers have been proposed to mitigate the ill-posed dilemma, while diverse deep networks have been recently built for the sake of performance improvement, please see Sec. 2 for details. However,

---

[*]Corresponding Author

[1]Many problems follow the same additive model, such as denoising ($I = B + N$, where $B$ and $N$ denote clean image and noise, respectively), and intrinsic image decomposition ($\log I = \log A + \log S$, where $A$ and $S$ stand for albedo and shading, respectively.) The proposed strategy can be potentially applied to all these tasks, but due to page limit, we concentrate on the task of SIRS to verify primary claims in this paper.

35th Conference on Neural Information Processing Systems (NeurIPS 2021).

in the literature, the additive property, say $I = T + R$, has been hardly investigated, except for acting as a reconstruction constraint.

Let us consider that, for any estimation pair $\hat{T}$ and $\hat{R}$ satisfying the additive property, there always exists an error/residual $Q$ subject to $I = \hat{T} + \hat{R} = (T + Q) + (R - Q)$. Once $Q$ is somehow obtained, the only thing needed to do is subtracting it from $\hat{T}$, and adding it into $\hat{R}$ instead of simply discarding. Under the circumstances, no information is trash, only misplaced. To be symmetric, we rewrite $Q$ as $Q := Q_T - Q_R$, yielding $T = \hat{T} - Q_T + Q_R$ and $R = \hat{R} - Q_R + Q_T$. In other words, the two targets including $\hat{T}$ and $\hat{R}$ can gain from each other by exchanging, rather than discarding, their respective "trash" factors $Q_T$ and $Q_R$. Driven by the above fact, a question naturally arises: *Can such an interaction/exchange be applied to intermediate deep features of dual-stream networks?*

**Contributions.** This paper answers the above question by designing a general interactive dual-stream/branch strategy, namely *your trash is my treasure* (YTMT). An obstacle to realizing YTMT was how to determine exchanging information. Intuitively, activation functions are competent for the job, which are developed to select (activate) a part of features from inputs. In this work, we adopt the ReLU that is arguably the most representative and widely-used activation manner, while others could be also qualified like [4, 24, 12, 2]. Please notice that, instead of simply discarding the deactivated features (trash) of the one stream, we alternatively deliver them to the other stream as compensation (treasure). By doing so, there are two main merits: 1) the information losing and dead ReLU problems can be consequently mitigated, and 2) the decreasing speed in training error can be significantly accelerated. The implementation is quite simple and flexible. We provide two optional YTMT blocks as examples and apply them on both plain and UNet architectures to verify the primary claims. Both ablation studies and experimental results on widely-used SIRS datasets are conducted to demonstrate the efficacy of YTMT, and reveal its superiority over other state-of-the-art alternatives.

## 2 Related Work

Over the past decades, much attention to resolving the image reflection separation problem has been drawn from the community. From the perspective of the required input amount, existing methods can be divided into two classes, *i.e.*, multi-image based and single image based methods.

**Multiple Image Reflection Separation (MIRS).** Utilizing multiple images [7, 28, 33, 27, 1, 8, 31, 22, 9, 29, 40, 32, 41, 10] used to be a prevalent way to cope with the task, as more information complementary to single superimposed images, like varied conditions and relative motions, can be explored from a sequence of images to accomplish the task. For instance, Agrawal *et al.* [1] use flash and no-flash image pairs to remove reflections and highlights, while Szeliski *et al.* [33] employ focused and defocused pairs. A variety of approaches [7, 27, 8, 22, 33, 40, 41] have been proposed to seek relationships between different images, among which Farid and Adelson [7] employ independent component analysis to reduce reflections and lighting. Gai *et al.* [8] alternatively develop an algorithm to estimate layer motions and linear mixing coefficients for the recovery. Li and Brown [22] adopt the SIFT-flow to align the images, while Xue *et al.* [40] enforce a heavy tailed distribution on the obstruction and background components to penalize the overlapped gradients, making the two layers independent. Moreover, Yang *et al.* [41] introduce a generalized double-layer brightness consistency constraint to connect optical flow estimation and layer separation. *Despite the satisfactory performance of MIRS methods, the need for specified shooting conditions and/or professional tools heavily limits their applicability.*

**Single Image Reflection Separation (SIRS).** In practice, single image based schemes are more attractive. No doubt that, compared with those MIRS techniques, single image based ones are also more challenging due to less information available, demanding extra priors to regularize the solution. To mitigate the ill-posedness, Levin *et al.* [19, 18] favor decompositions that have fewer edges and corners by imposing sparse gradients on the layers, in the same spirit as [40]. Levin and Weiss [17] allow users to manually annotate some dominant edges for the layers as explicit constraints for the problem, which requires careful human efforts to obtain favorable results. Li and Brown [23] assume one layer is smoother than the other, and penalize differently on the two layers in the gradient domain to split the reflection and transmission. *Despite a progress made toward addressing the problem, the assumption and requirement could be frequently violated in real situations. Besides, these methods all rely on handcrafted features that may considerably restrict their performance.*

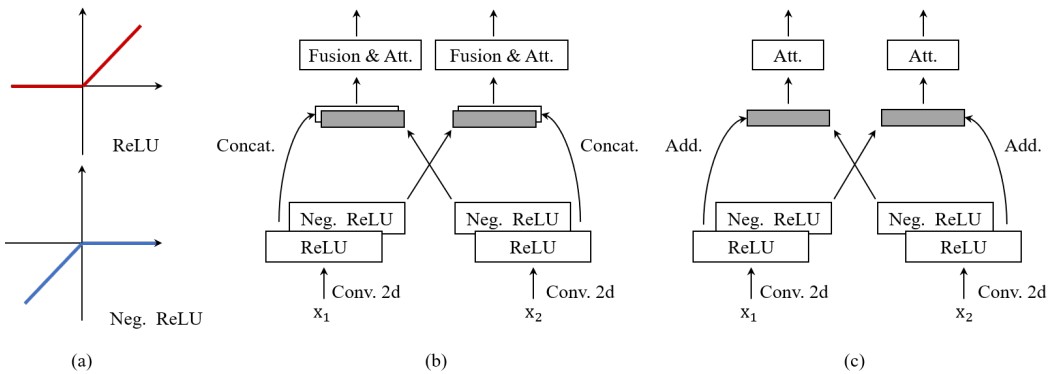

Figure 1: The functional behaviors of ReLU and negative ReLU are plotted in (a). Two YTMT block options are given in (b) and (c). To fuse features from the normal and YTMT connections, the first option uses channel concatenation and $1 \times 1$ convolutions, while the second simply uses feature addition. Pixel and channel attention mechanism (denoted by Att.) is utilized in both (b) and (c).

With the emergence of deep neural networks, learning based methods [6, 44, 42, 38, 20, 39] have shown their power and become dominant for the task. Concretely, CEILNet [6] comprises of two stages, similarly to [17, 37]. An edge map is estimated in the first stage, which performs as the guidance for the second stage to produce the final transmission layer. However, the network is hard to capture high-level information, thus having trouble guaranteeing the perceptual quality. Zhang *et al.* [44] further take the perception cue into consideration by coordinating hyper-column features [11], the perceptual and adversarial losses. The exclusivity loss (see also the gradient production penalization in [40]) is designed for assuring gradients to be exclusive between the two components. This manner is also adopted by [38]. The main drawback is the insufficient capacity in processing cases with large transmission-reflection overlapped regions. Recently, ERRNet [38] was proposed, which enlarges the receptive field by introducing channel attention and pyramid pooling modules. It does not explicitly predict the reflection layer, which directly discards possible benefit from the supervision of reflection. IBCLN [20] proposes an iterative boost convolutional LSTM network to progressively split the two layers. However, such an iterative scheme slows down the training and predicting procedure. The errors will also be accumulated due to the high dependency on the outputs of previous stages. Though the learning based strategies have further stepped forward in single image reflection separation compared with the traditional methods, they either restore the target layers individually, or together with some concerns dealing with outputs (*e.g.*, enforcing the linear combination constraint and/or concatenating them together as input). In other words, *they barely take into account the interaction across the two streams/branches, which is key to the target task and also other dual-stream decomposition tasks.* This study is mainly to demonstrate the effectiveness of such an interaction consideration.

## 3 Deep Interactive Dual-Stream Learning for SIRS

### 3.1 YTMT Strategy

In this part, the proposed deep interactive dual-stream strategy will be detailed by centering around the concept that your trash is my treasure, which would be beneficial to a variety of two-component decomposition tasks using dual-branch networks. Prior works [21, 35] have shown evidence on the effectiveness of passing information between two branches though, our YTMT strategy performs in a novel and principled way. We first give the definition of the **negative ReLU** function as follows:

$$\text{ReLU}^-(\mathbf{x}) := \mathbf{x} - \text{ReLU}(\mathbf{x}) = \min(\mathbf{x}, 0), \tag{1}$$

where $\text{ReLU}(\mathbf{x}) := \max(\mathbf{x}, 0)$. By the negative ReLU, the deactivated features can be easily retained. Figure 1 (a) exhibits the behaviors of the ReLU and negative ReLU.

Here, let $\mathbf{x}^0$ be the input to the first layer of the network, and $\tilde{\mathbf{x}}_i^l$ ($i \in \{1, 2\}$ for two branches) denotes the feature obtained by the $i$-th branch after $l$ stacked layers, *i.e.* $\tilde{\mathbf{x}}_i^l := \mathcal{H}_i^l(\mathbf{x}^0)$. The inputs to the

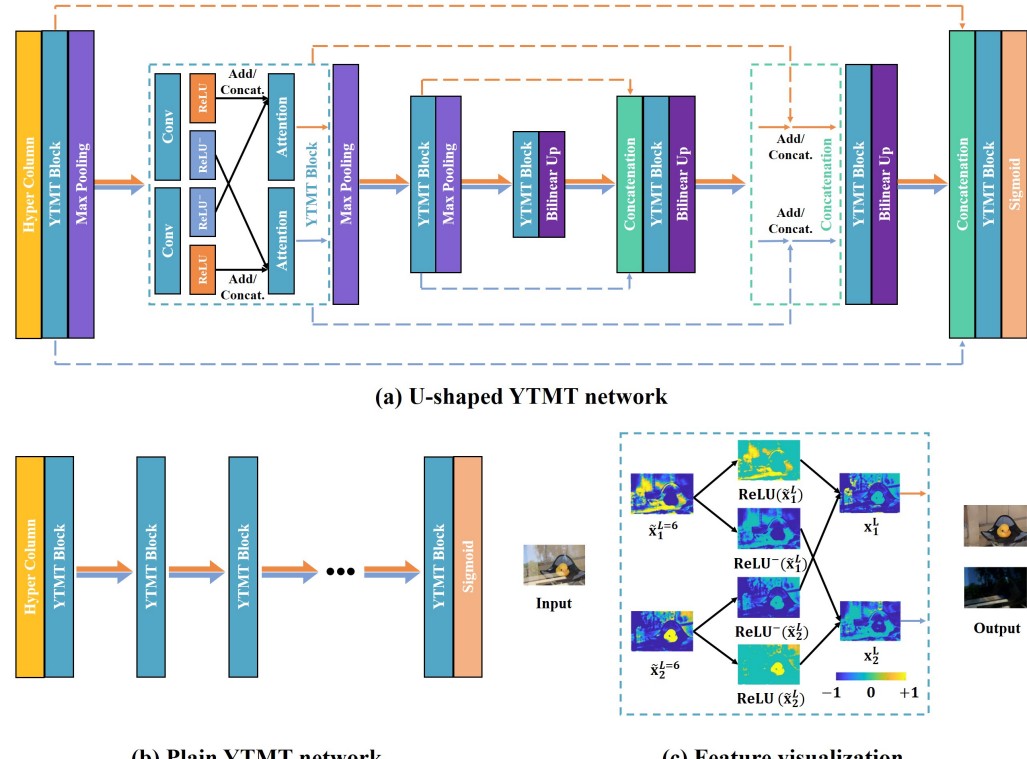

(a) U-shaped YTMT network

(b) Plain YTMT network

(c) Feature visualization

Figure 2: Illustration of the YTMT networks. (a) and (b) offer the YTMT versions modified on the U-shape and plain network architectures, respectively. The input is first augmented by hyper column [44] and then fed into the YTMT blocks. (c) shows a visualized example of producing $\mathbf{x}_1^L$ and $\mathbf{x}_2^L$ from $\tilde{\mathbf{x}}_1^L$ and $\tilde{\mathbf{x}}_2^L$ at the sixth YTMT block of the trained U-shaped YTMT network. For better view, all the features are normalized by softmax.

$(l+1)$-th layer are as follows:

$$\mathbf{x}_1^l := \mathrm{ReLU}(\tilde{\mathbf{x}}_1^l) \oplus \mathrm{ReLU}^-(\tilde{\mathbf{x}}_2^l); \quad \mathbf{x}_2^l := \mathrm{ReLU}(\tilde{\mathbf{x}}_2^l) \oplus \mathrm{ReLU}^-(\tilde{\mathbf{x}}_1^l), \tag{2}$$

where $\oplus$ can be either the concatenation operation or the addition between features activated by the $\mathrm{ReLU}$ function (called **normal connection**) and those by the $\mathrm{ReLU}^-$ (**YTMT connection**), as shown in Fig. 1 (b) and (c). As can be obtained from Eq. (2), the amount of information in $\mathbf{x}_1^l$ and $\mathbf{x}_2^l$ is equivalent to that in $\tilde{\mathbf{x}}_1^l$ and $\tilde{\mathbf{x}}_2^l$. This property guarantees no information flowing away from the interaction, which substantially avoids the problems of vanishing/exploding gradients and dead ReLU. Figure 2 (c) depicts a visual example of producing $\mathbf{x}_1^L$ and $\mathbf{x}_2^L$ from $\tilde{\mathbf{x}}_1^L$ and $\tilde{\mathbf{x}}_2^L$ (in this case, $L = 6$). It shows that $\mathrm{ReLU}^-(\tilde{\mathbf{x}}_1^L)$ is complementary to the $\mathrm{ReLU}(\tilde{\mathbf{x}}_2^L)$ and $\mathrm{ReLU}^-(\tilde{\mathbf{x}}_2^L)$ is complementary to the $\mathrm{ReLU}(\tilde{\mathbf{x}}_1^L)$. By merging the complementary counterparts, there is no information wasted by the rectifiers. In addition, this strategy can significantly speed up the decreasing of training error similarly to the ResNet design [13], which will be empirically validated in Sec. 4. Moreover, generally speaking, our strategy can be compatible with most, if not all, of the activation pairs (e.g. Softmax and Softmin). But for the additive problems, the pair of ReLU and negative ReLU is more suitable due to its "either A or B" nature that satisfies the task of SIRS.

We again emphasize that the proposed YTMT strategy is general and flexible, which can be implemented in various forms according to different demands. Figure 1 provides two YTMT block options. The one in (b) fuses features from the normal and YTMT connections by channel concatenation and $1 \times 1$ convolutions, while the second in (c) simply employs feature addition. In addition, pixel and channel attention mechanism is introduced in both (b) and (c) to select and re-weight the merged features (see [25] for details). Moreover, the YTMT blocks can be applied to most, if not all, of dual-stream backbones by simple modification. In Fig. 2 (a) and (b), two commonly-used architectures,

*i.e.* U-shaped [26] and plain [43] networks, are present. We will shortly show how to construct these YTMT based networks, and demonstrate their improvement over the backbones and the superior performance on specific applications over other competitors.

## 3.2 YTMT based Design for SIRS

As shown in Fig. 2 (a), we adopt the U-shaped network as the backbone for the task of SIRS, which can be readily implemented by replacing the convolutional blocks in the UNet architecture with proposed YTMT block options. Following the prior work, the input images are first augmented by the hypercolumn [44], gaining 1475 channels, then mapped to 64 via a $1 \times 1$ convolution to fuse VGG features and the original input. Each YTMT block is an interactive dual-stream module containing two convolutional layers, both followed by a dual-ReLU rectifier pair. The activations produced by the negative ReLU rectifiers are exchanged between the two streams, then merged by feature addition or concatenation operator before being fed into the attention block. We here use max-pooling and bilinear interpolation to squeeze and expand the feature maps. Like in the single-stream UNet, there are skip connections between the encoder layers and the decoder layers (represented by dashed arrows in orange in Fig. 2 (a)), but an extra skip connection is added between each encoder-decoder layer pair for the dual-stream design (represented by dashed arrows in blue). The features in skip connections are first fused with the up-sampled features and then fed into the YTMT blocks. After training the first stage to converge, we then freeze its parameters and initialize the second stage with them. The outputs of the first stage are then fed into the second one for training until it also converges. Under this design, the information of the transmission and reflection pairs will further interact during passing through the second stage. To be concluded, with consideration of YTMT block options ("C" for channel concatenation and "A" for feature addition) and using two-stage or not ("S" for single-stage and "T" for two-stage), YTMT based design for SIRS has several variants, including YTMT-{UCS, UAS, UCT, UAT}, based on the choice of model architecture ("U" stands for the U-shaped architecture. We omit the plain architecture for its inferior efficiency compared with the U-shaped on the task of SIRS). The performance difference between these YTMT variants will be studied in Sec. 4.3.

The objective function considered for SIRS consists of reconstruction loss, perceptual loss, exclusion loss, and adversarial loss. In what follows, each term is explained.

**Reconstruction loss.** Preceding methods have revealed that edges are essential for a valid separation [17, 40, 6, 35]. To make our model sensitive to the gradients, we follow [6, 38] to penalize the gradient difference between predictions and targets besides the MSE term, via:

$$\mathcal{L}_{rec} := a\|\hat{T} - T\|_2^2 + b\|\hat{R} - R\|_2^2 + c\|\nabla\hat{T} - \nabla T\|_1 + d\|\hat{T} + \hat{R} - I\|_1, \tag{3}$$

where $\|\cdot\|_1$ stands for the $\ell_1$ norm, and $\|\cdot\|_2$ the $\ell_2$ norm. In addition, we empirically set $a = 0.3$, $b = 0.9$, $c = 0.6$, $d = 0.2$ in all of our experiments. Since reflections are usually weak, the penalization on $\nabla\hat{R} - \nabla R$ is omitted for stable training.

**Perceptual Loss.** The perceptual loss [14] assists models in achieving high perceptual quality. We minimize the $\ell_1$ difference between the features of predicted components and those of ground-truths at layers 'conv2_2', 'conv3_2', 'conv4_2', and 'conv5_2' of a VGG-19 model [30] pretrained on the ImageNet dataset [3]. Denoting the features of the input at layer $i$ as $\phi_i(\cdot)$, we have the following function:

$$\mathcal{L}_{per} := \sum_j \omega_j \|\phi_j(T) - \phi_j(\hat{T})\|_1 + \sum_j \omega_j \|\phi_j(R) - \phi_j(\hat{R})\|_1, \tag{4}$$

where $\omega_j$s balance the weights of different layers, $\{0.38, 0.21, 0.27, 0.18, 6.67\}$ as default.

**Exclusion Loss.** Prior work [40, 44] shows that enforcing the exclusivity on the two components in the gradient domain is beneficial to separation tasks, which is defined as:

$$\mathcal{L}_{exc} := \frac{1}{N} \sum_{n=0}^{N-1} \|\Psi(\hat{T}^{\downarrow n}, \hat{R}^{\downarrow n})\|_2^2 \quad \text{with} \quad \Psi(\hat{T}, \hat{R}) := \tanh\left(\lambda_T |\nabla\hat{T}|\right) \circ \tanh\left(\lambda_R |\nabla\hat{R}|\right), \tag{5}$$

where $\hat{T}^{\downarrow n}$ and $\hat{R}^{\downarrow n}$ are $\hat{T}$ and $\hat{R}$ down-sampled by $2^n$ times. In addition, $\lambda_T$ and $\lambda_R$ are normalization factors and $\circ$ represents element-wise multiplication.

Table 1: Quantitative results on four real-world benchmark datasets of different methods. The best results are indicated in red and the second best results in blue.

| Datasets | Metrics | Input | CEILNet | Zhang *et al.* | BDN | ERRNet | IBCLN | Lei *et al.* | YTMT-UCT |
|---|---|---|---|---|---|---|---|---|---|
| Real20 (20) | PSNR | 19.16 | 18.45 | 22.55 | 18.41 | 22.89 | 21.86 | 22.35 | 23.26 |
| | SSIM | 0.732 | 0.690 | 0.788 | 0.726 | 0.803 | 0.762 | 0.793 | 0.806 |
| Objects (200) | PSNR | 23.74 | 23.62 | 22.68 | 22.72 | 24.87 | 24.87 | 23.81 | 24.87 |
| | SSIM | 0.878 | 0.867 | 0.879 | 0.856 | 0.896 | 0.893 | 0.882 | 0.896 |
| Postcard (199) | PSNR | 21.31 | 21.24 | 16.81 | 20.71 | 22.04 | 23.39 | 21.48 | 22.91 |
| | SSIM | 0.877 | 0.834 | 0.797 | 0.859 | 0.876 | 0.875 | 0.873 | 0.884 |
| Wild (55) | PSNR | 26.06 | 22.36 | 21.52 | 22.36 | 24.25 | 24.71 | 23.84 | 25.48 |
| | SSIM | 0.890 | 0.821 | 0.832 | 0.830 | 0.853 | 0.886 | 0.866 | 0.890 |

Table 2: Quantitative comparison in reflection recovery between the top-3 methods in transmission recovery. The best results are highlighted in red and the second best results in blue.

| Datasets | Metrics | ERRNet | IBCLN | YTMT-UCT |
|---|---|---|---|---|
| Real20 (20) | PSNR | 23.55 | 22.36 | 24.30 |
| | SSIM | 0.446 | 0.469 | 0.542 |
| Objects (200) | PSNR | 26.02 | 19.97 | 26.45 |
| | SSIM | 0.446 | 0.226 | 0.499 |
| Postcard (199) | PSNR | 22.47 | 13.16 | 23.41 |
| | SSIM | 0.419 | 0.230 | 0.478 |
| Wild (55) | PSNR | 25.52 | 20.83 | 27.33 |
| | SSIM | 0.460 | 0.298 | 0.590 |

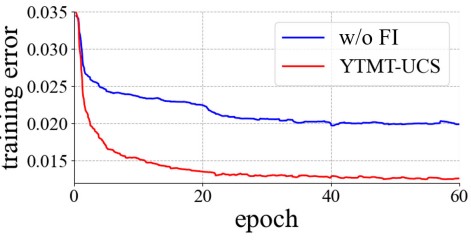

Figure 3: Training error on our training data of YTMT-UCS and a dual-stream UNet without any feature interaction (w/o FI). The proposed interaction strategy leads to a much faster decreasing speed.

**Adversarial Loss.** The adversarial loss regulates the SIRS solution on the real-world image manifold. With the adversarial training of the main network $\mathcal{G}$ and a discriminator $\mathcal{D}$, they seek the equilibrium by optimizing the adversarial loss $\mathcal{L}_{adv}$ [15], as follows:

$$\mathcal{L}_{adv}^{G} := -\log(\mathcal{D}(T, \hat{T})) - \log(1 - \mathcal{D}(\hat{T}, T)); \quad \mathcal{L}_{adv}^{D} := -\log(1 - \mathcal{D}(T, \hat{T})) - \log(\mathcal{D}(\hat{T}, T)), \quad (6)$$

where $\mathcal{D}$ stays invariant with [38].

Our overall objective function turns out to be:

$$\mathcal{L}_{tot} := \mathcal{L}_{rec} + \lambda_1 \mathcal{L}_{per} + \lambda_2 \mathcal{L}_{exc} + \lambda_3 \mathcal{L}_{adv}, \quad (7)$$

with $\lambda_1 = 0.1$, $\lambda_3 = 1$, and $\lambda_3 = 0.01$ empirically set.

## 4 Experimental Validation on SIRS

### 4.1 Implementation Details

Our training data consists of both real-world and synthesis images, as used in [38]. Among these data, 90 pairs of input and transmission groundtruth are collected by Zhang *et al.* [44], and 7,643 image pairs chosen from the PASCAL VOC dataset [5] to synthesize superimposed images following CEILNet [6]. The models are implemented in PyTorch and optimized with Adam optimizer, keeping $\beta_1 = 0.9$, and $\beta_2 = 0.999$. The learning rate is initialized as $10^{-4}$ and then reduced by half at epoch 60, 80, and 100, respectively. The training is stopped at epoch 120. All the models are trained on a single RTX 2080 Ti graphics card.

### 4.2 Comparison with State-of-the-art Methods

We select the proposed YTMT-UCT network to compare with the state-of-the-art methods, including Zhang *et al.* [44], BDN [42], ERRNet [38], IBCLN [20] and Lei *et al.*[16], on four real-world dataset, involving Real20 [44] and three subsets of SIR$^2$ [34]. The PSNR and SSIM metrics are utilized to evaluate all the competing methods as shown in Table 1. Generally speaking, each component for blind source separation tasks should be treated equally. For the SIRS task, the reflection layer

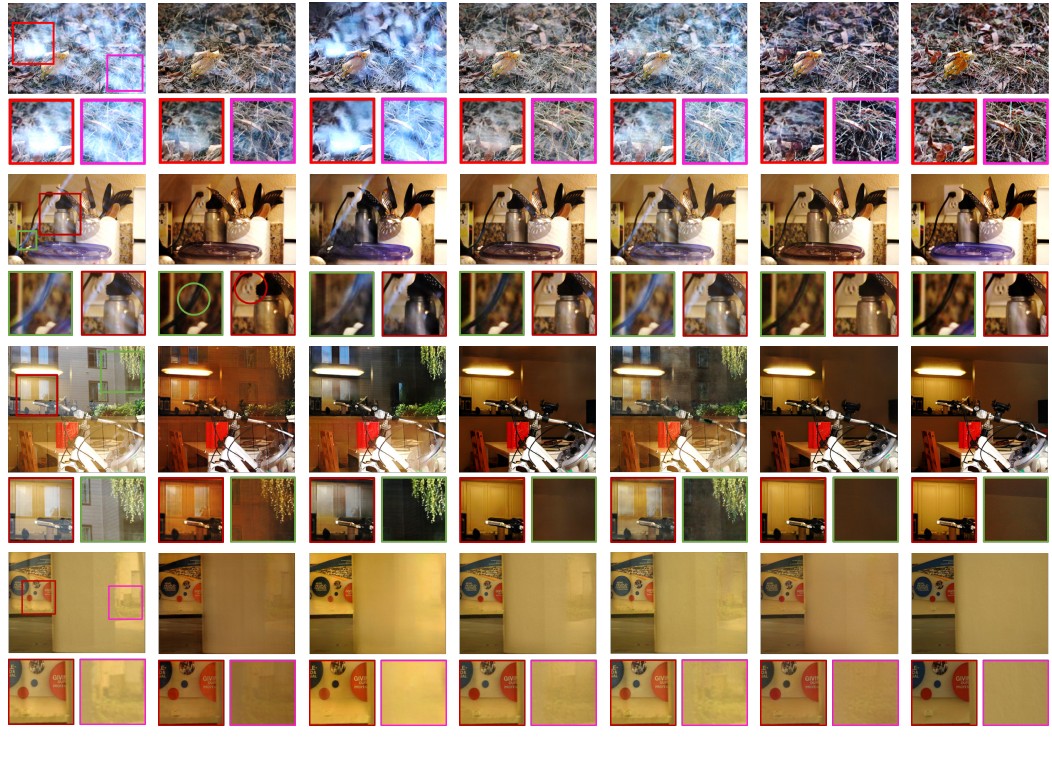

| Input | Zhang *et al.* [44] | BDN [42] | ERRNet [38] | IBCLN [20] | Ours | GT |

Figure 4: Visual comparison of state-of-the-art methods and ours on real-world testing dataset, including Real20 [44] and SIR$^2$ [6]. It shows that the results by our method are more visually striking compared with those by the competitors, with less artifacts and color distortion (please pay attention to the regions framed by red boxes).

Table 3: Quantitative results on the testing samples from Nature dataset of different methods. The best results are highlighted in red and the second best results in blue.

| Metrics | CEILNet-F | Zhang *et al.* | BDN-F | RmNet | ERRNet-F | IBCLN | YTMT-UCS |
|---------|-----------|----------------|-------|-------|----------|-------|----------|
| PSNR | 19.33 | 19.56 | 18.92 | 19.36 | 22.18 | 23.57 | 23.85 |
| SSIM | 0.745 | 0.736 | 0.737 | 0.725 | 0.756 | 0.783 | 0.810 |

can be a critical part of the information to reproduce the image shooting scene, as discussed in [36]. Therefore, we report the performance difference of the reflection recovery in Table 2, among the methods with the top-3 transmission recovery. The visual comparison is also conducted in Fig. 6. Given ERRNet does not output reflection layers and the predicted reflections of IBCLN still mingle with large parts of transmission layers, we compare their reflection layers by $I - \hat{T}$ for fairness.

It turns out that the YTMT-UCT achieves the best results on almost all of the testing datasets, in terms of the recovery of both transmission and the reflection layers, which indicates that the YTMT strategy gives the most accurate separations with the help of frequent feature interaction between the transmission and reflection streams. The difficulty raising in Real20 comes from the largely overlapped regions and various reflection patterns, while the challenge of SIR$^2$ is mainly low-quality transmission layers and reflections blended with the monochromatic background. Therefore, we further conduct the qualitative comparison among these methods in Fig. 4 to show our capacity for these challenging samples. As can be observed, the method proposed by Zhang *et al.* produces results with severe color distortion and cannot handle the aforementioned globally overlapping problem. BDN and IBCLN generalize unsatisfyingly on the Real20 dataset, failing to remove conspicuous reflections in several cases, and some reflections are even enhanced by the BDN due to the inaccurate reflection estimation amplified by the cascade structure. Meanwhile, both the methods refine the

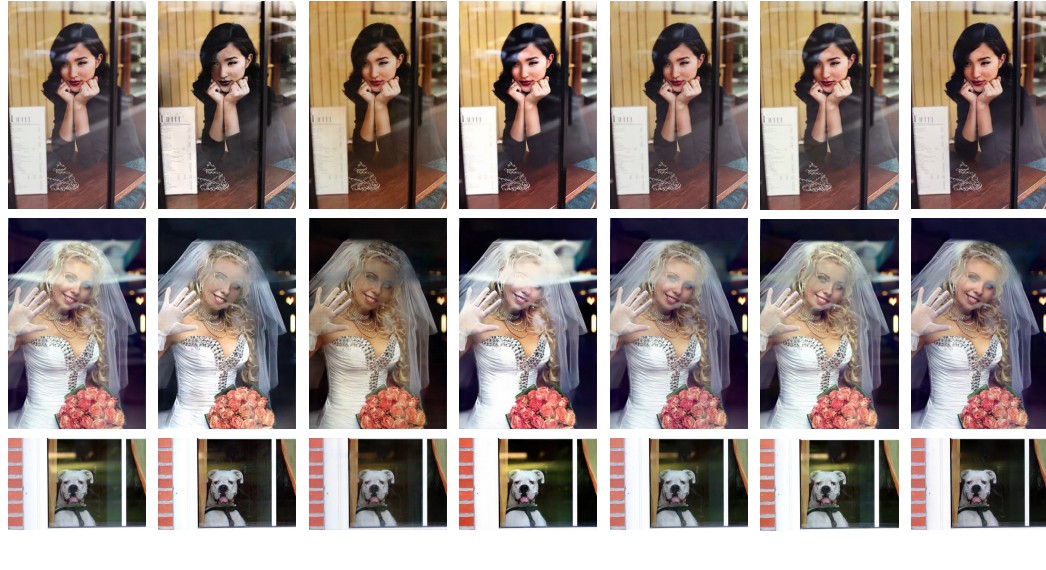

| Input | CEILNet [6] | Zhang *et al.* [44] | BDN [42] | ERRNet [38] | IBCLN [20] | Ours |

Figure 5: Visual comparison of YTMT-UAT and state-of-the-art methods on real45 dataset[6]. It shows that our method produces results with fewer residual reflections.

separations stage-by-stage or iteration-by-iteration, where new outputs are concatenated to be fed into the next stage/iteration, instead of inner interaction or feature exchanging. As a consequence, the error would be rather accumulated. However, in the YTMT strategy, there are more chances for the network (e.g., 14 times in the YTMT-UCT). Moreover, the separations are estimated in a single network, rather than several sub-networks. We train the first stage of the network to converge, and then use it to initialize the second stage. The training of the second stage depends on not only the preliminary separations, but also the knowledge learned in the first stage. If there is no better separation the network can exploit, the second stage will produce a solution close to the preliminary separations. ERRNet introduces some useful network building blocks, but still leaves some obvious reflections. It merely estimates the transmission layer, discarding the interaction between two components. In conclusion, feature interaction is highly desired for the satisfactory separations. Moreover, the training error shown in Fig. 3 further accounts for the information efficacy of YTMT strategy, given the large margin of the convergence between the dual-stream UNet without any feature interaction and the YTMT-UCS solution. To evaluate our generalization ability to different data distributions, we also follow the training setting of IBCLN [20] and report our results on the testing data of the Nature dataset proposed by it. We achieve 23.85 and 0.810 for PSNR and SSIM, respectively, compared with 23.57 and 0.783 of IBCLN, and surpass previous methods more than a lot as shown in Table 3. Note that we reuse the numerical results proposed in the paper of IBCLN, since the same experimental setup is kept.

### 4.3 Ablation Study

To further assess the capacity of the YTMT strategy for the task, we have developed two baselines to illustrate the effectiveness of our proposed YTMT strategy (see results in Table 4). The first model is a dual-stream UNet without any feature interaction. One might think it is identical to a single UNet. However, constrained by the reconstruction loss term $\|\hat{T} + \hat{R} - I\|_1$, the estimations of the two layers $\hat{T}$ and $\hat{R}$ are highly related to each other. Moreover, supervised by the ground-truth of both transmission and reflection layers, the optimization constrained by multiple regularizers are more likely to improve the generalization ability of the network. Thus we adopt it as one of the baseline networks, which means any interactive method should not have lower performance than it. The second model replaces all the negative ReLU rectifiers by the normal ReLU in the YTMT-UCS to validate if the YTMT strategy really works to preserve the discarded information. The rest four models are the YTMT variants introduced in the Sec. 3.2.

Table 4: Ablation study on different network architectures, including a dual-stream UNet without any feature interaction (FI), a YTMT-UCS with all the negative ReLU rectifiers replaced by the normal ReLU and four YTMT variants. The best results are highlighted in red and the second best results in blue.

| Datasets | Metrics | w/o FI | ReLU only | YTMT-UCS | YTMT-UCT | YTMT-UAS | YTMT-UAT |
|----------|---------|--------|-----------|----------|----------|----------|----------|
| Real20 | PSNR | 20.57 | 22.79 | 23.05 | 23.09 | 23.26 | 23.39 |
|  | SSIM | 0.752 | 0.802 | 0.805 | 0.802 | 0.801 | 0.809 |
| Objects | PSNR | 23.85 | 24.06 | 24.46 | 24.58 | 24.71 | 25.40 |
|  | SSIM | 0.883 | 0.886 | 0.891 | 0.891 | 0.893 | 0.899 |
| Postcard | PSNR | 20.82 | 22.02 | 22.66 | 22.75 | 22.45 | 23.01 |
|  | SSIM | 0.863 | 0.869 | 0.885 | 0.884 | 0.871 | 0.874 |
| Wild | PSNR | 24.20 | 24.71 | 25.24 | 25.46 | 24.49 | 25.19 |
|  | SSIM | 0.881 | 0.859 | 0.887 | 0.892 | 0.870 | 0.880 |

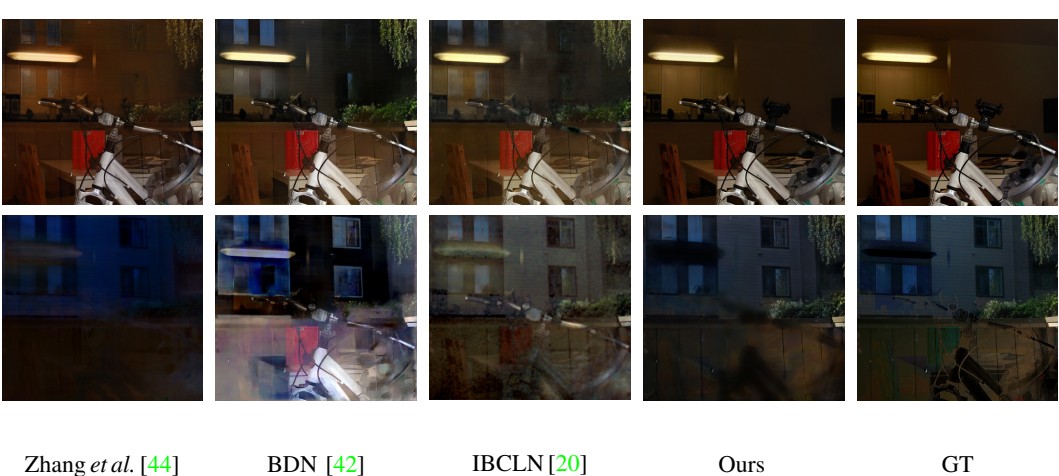

Zhang *et al.* [44]      BDN [42]      IBCLN [20]      Ours      GT

Figure 6: Visual comparison on the layer decomposition between YTMT-UCT and state-of-the-art methods that output transmission-reflection layer pairs.

With approximate number of parameters, YTMT solutions show obvious superiority over other alternatives, benefiting from the strategy introducing the complementary interaction between dual streams. Obviously, without any feature interaction, a dual-stream UNet is apparently inferior to those with feature interaction, which partly explains the performance gap between the IBCLN and ours. With only the positive ReLU rectifiers, the dual streams in the network will have the same activations, leading to degraded performance. Moreover, it can be seen that the YTMT-UAT shows leading results on the first two datasets, while YTMT-UCT exhibits better overall performance over other state-of-the-art methods, thus it is opted as the default architecture for SIRS tasks. Also, it shows that a two-stage architecture can indeed further improve the results.

## 5 Conclusion

In this paper, we proposed a general rule for deep interactive dual-stream/branch learning, namely your trash is my treasure (denoted as YTMT), which says that two branches should communicate with each other frequently by exchanging, rather than discarding, the information useless to themselves. Activation functions are deemed to be suitable for determining which part of information to exchange. As an example, the widely-used ReLU was chosen to validate the primary claims of this work. In addition, we offered several YTMT based designs based on both U-shaped and plain backbones to show the flexibility. Extensive experimental results on public SIRS datasets have been provided to verify the effectiveness of YTMT, and demonstrate the clear advantages of our method in comparison with other state-of-the-art alternatives. Another merit of YTMT comes from its acceleration in decreasing error during training. It is positive that our strategy can derive diverse designs and be beneficial to many two-component decomposition tasks.

## Acknowledgement

This work was supported by the National Natural Science Foundation of China under Grant nos. 62072327 and 61772512, and TSTC under Grant no. 20JCQNJC01510.

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
