# OpenReview forum: "Trash or Treasure? An Interactive Dual-Stream Strategy for Single Image Reflection Separation"
_NeurIPS.cc/2021/Conference — NeurIPS 2021 Poster_

### Official Review · Reviewer_SZth · 2021-07-07

**Rating:** 6
**Confidence:** 4

**Summary:**


This paper presents a method that recovers reflection and transmission components from a given image degraded with reflections. The proposed network employs a dual-stream interactive strategy where the discarded information (by ReLU) from one branch helps improving the feature representation of the other branch and vice versa. Experiments are performed for several image processing tasks, including image reflection removal, denoising, demoireing, and intrinsic image decomposition.


**Limitations And Societal Impact:**

Adequately addressed.

**Main Review:**

The idea is simple and provides favorable results for image decomposition tasks, especially for image reflection removal.  However, I have the following concerns.


* Image reflection removal is a position-sensitive task where pixel-to-pixel correspondence from the input to output images is required. For such restoration tasks, those methods that operate at high-resolution (having no downsampling operations) usually perform better than the UNet-like architectures. Why did you use YTMT-UCT as a default choice over the YTMT-PCT version?



* Among the UNet versions of YTMT, Why do you opt for YTMT-UCT as a default architecture to perform main experiments, when in the ablation experiments (Table 4) YTMT-UAT shows overall better performance than the YTMT-UCT version?



* The evaluation for the image denoising task is not convincing. The proposed method performs poorly when compared to DnCNN (which is five years old method) for Gaussian noise levels up to sigma=75. The difference in performance might increase even further when compared to recent image denoising approaches such as RIDNet, RDN, PAN, etc. It is only when the noise levels are sigma=80 and above, the proposed method shows performance improvement over DnCNN. Don't you think that the experimental setting with such high noise levels is unrealistic and impractical?



* Since the evaluation for the image reflection removal task is performed on real-world datasets, it would have been interesting to see the performance of YTMT on \textit{blind} image denoising task on real datasets such as SIDD and DND.



* In Table 1 you are comparing the results produced by the proposed YTMT method with those of the IBCLN algorithm (CVPR20). The authors of IBCLN also provided a new real-world dataset called \textit{Nature}. Did you compute the image quality scores on the Nature dataset?



* The main architecture diagram (Figure 2a) is complicated and the text explaining it isn't well written either (Sec 3.2). While some parts are fine, the overall paper is not easy to follow.



 * * * *  *  *  *  *

### **Post Rebuttal Comments**
* The authors have provided a convincing response to my concerns. Overall, the idea of this paper is simple and intuitive and provides good results. I recommend this paper for acceptance. However, I would suggest the authors to revise the paper to improve the text and simplify the figures.

**Time Spent Reviewing:**

~ 24 hours

---

> ### Author Response · Authors · 2021-08-10
> **Responses to the comments from reviewer SZth**
>
> First of all, thanks for your positive feedback on the novelty and potential of this work. We provide the responses to the main concerns from the reviewer as follows.
>
> **Q1**: Why use YTMT-UCT as a default choice over the YTMT-PCT version?
>
> **A1**: We have not observed any relevant issues about the position-sensitive problem during our experiments for the YTMT-UCT. The UNet-like architectures naturally introduce the multi-scale information, which is essential for the samples with large overlap regions. Moreover, YTMT-Us provide faster training/inference speed. When given a specific amount of parameters (e.g. the same amount as YTMT-UCS), YTMT-PCS takes fairly more computational sources (memory and computing) to train. Experimentally, we find that YTMT-UCS can offer satisfactory separations for SIRS with less computational cost.
>
> **Q2**: Why adopt YTMT-UCT as a default architecture to perform main experiments over YTMT-UAT?
>
> **A2**: Despite more favorable results on real20 and objects datasets, YTMT-UAT does not surpass some of the metrics made by other methods on postcard and wild datasets for the current stage of experiments. However, if users want to utilize YTMT networks in the scenarios whose distribution is more akin to real20 and objects, the YTMT-UAT is a better option.
>
> **Q3**: Should the YTMT networks be compared with more recent state-of-the-art image denoising approaches and applied on real denoising datasets?
>
> **A3**: We adopt DnCNN as a baseline because of its simple but effective network architecture design, for the sake of focusing on the YTMT strategy. We compare our strategy with DnCNN both in pure settings, which can more explicitly illustrate the generalization ability of our strategy. It is important given that the noise does not always fall within the training range in the real world. In our experiment of Section 5, no extra modules except for the YTMT blocks are introduced, and the number of parameters and the depth of the network are even disadvantaged compared with the DnCNN network.  Moreover, the YTMT network has to restore both the natural images and the noises, which means there will be fewer parameters used to fit the natural images. However, the difference between YTMT-PCS and DnCNN is trivial and DnCNN-2 is surpassed, which verifies the gain of the information exchange. Given lots of the state-of-the-art image denoising methods are utilizing a single branch residual learning scheme as $\hat{I} = I + f(I)$, where $I$ is the input image, $f(\cdot)$ is a deep model and $\hat{I}$ is the final output, the deep model is actually fitting the noise, and the catastrophic decrease of their performance like the DnCNN can be expected while the noise level goes higher. Therefore, our YTMT strategy has the potential to be a much more robust possible solution, regularizing the solution space from both directions of natural images and noises, and exchanging their respective information in the feature space. Still, more complicated experiments are out of the scope of this paper, and we only give it a brief glance in Section 5.
>
> This paper mainly focuses on the effectiveness of the YTMT strategy on the specific SIRS task for now, and it achieves state-of-the-art performance on most of the testsets. We also discuss its potential flexibility for wider usage, but developing a powerful denoising network against the recent state-of-the-art methods is not the purpose of this paper. It is clear that more task-specific modules can be introduced to reach higher performance, but it will divorce the focus of this paper. A noteworthy interesting point is that our YTMT-UCS (w/o any extra design) can easily achieve competitive results with the state-of-the-art method (MBCNN, task-specific modules like learnable bandpass filter are utilized) and surpass other methods by a large margin. One can refer to Section C of the supplementary material if interested.
>
> **Q4**: Compute the image quality scores on the Nature dataset?
>
> **A4**: To keep similar training data settings, we train our model on 2800 synthetic images randomly selected from the synthetic dataset of Zhang et al. and 290 real-world training images, which are the same as being used in IBCLN. We achieve 23.85 and 0.810 for PSNR and SSIM, respectively, compared with 23.57 and 0.783 of IBCLN, and surpass previous methods more than a lot as shown below. This can further demonstrate the superiority of the proposed YTMT strategy on the SIRS task.
>
> | Dataset    | Metrics | CEILNet-F | Zhang et al. | BDN-F | RmNet | ERRNet-F | IBCLN | YTMT-UCS |
> |------------|---------|-----------|--------------|-------|-------|----------|-------|----------|
> | Nature(20) | PSNR    | 19.33     | 19.56        | 18.92 | 19.36 | 22.18    | 23.57 | **23.85**    |
> |            | SSIM    | 0.745     | 0.736        | 0.737 | 0.725 | 0.756    | 0.783 | **0.810**    |
>
> **Q5**: Explanation of the main architecture diagram?
>
> **A5**: Briefly speaking, Fig. 2 (a) is two UNets with their ReLU rectifiers replaced by the ReLU and negative ReLU pairs. They exchange their information through swapping their features activated by the negative ReLU rectifiers. Note that we not only exchange their information between adjacent blocks but also exchange through the skip connections. The technical details can be found in our code submitted as the supplementary material. (c) is the feature visualization of what happens during the information interaction. (b) is the architecture of YTMT-P networks.
>
> We sincerely appreciate your suggestions, and will definitely proofread and revise the paper according to the comments to improve its quality. In addition, we will put more words on the architecture for clarity.

---

### Official Review · Reviewer_myi5 · 2021-07-11

**Rating:** 7
**Confidence:** 4

**Summary:**

This paper presents a nonlinear module that leverages both positive and negative ReLUs for feature separation. This enables applications such as reflection removal, blind denoising and other layer separation tasks. The paper shows several variants of adding the proposed module into existing CNN frameworks. Results show superior performance on several benchmark datasets.

**Ethical Concerns:**

Not that I'm aware of.

**Limitations And Societal Impact:**

**Missing references**

The paper is missing a set of works on internal learning for layer separation:
- Ulyanov, Dmitry, Andrea Vedaldi, and Victor Lempitsky. "Deep image prior." CVPR 2018.
- Gandelsman, Yosef, Assaf Shocher, and Michal Irani. "" Double-DIP": Unsupervised Image Decomposition via Coupled Deep-Image-Priors." CVPR 2019.

**Writing**

Writing is mostly clear.
- line 164 & 238: the choose of -> the choice of

**Main Review:**

The idea of this paper is simple and intuitive. It's executed well with promising results and good ablation study. The novelty of this paper is the way of exploiting all features learned in a CNN and identifying its effectiveness in layer separation tasks where pixels are interfered but useful. The training scheme, data generation, objective functions are all from prior works, but the simple design of keeping all features in the network by adding dual-ReLU branch is interesting and novel, and can potentially inspire further research works.

Nevertheless, I have the following concerns:

**Results**

Although the model is tested on several datasets, not all results are shown in the supplement and the paper. Quality degrades (compared to results in the paper) for some images in fig 4 and 5 in the supplement. The limitation of the relatively worse performance isn't clear.

**Ablation**

While the paper presents several ablation studies on how features are fused and the backbone architectures, some supporting experiments are missing for the key claim of keeping the 'trash' features. In other words, is the benefit from having the negative branch?

What happens if the model only keeps the negative ReLU or only the positive ReLU (which is similar to the vanilla backbone)? For example, to keep the number of parameters the same, how does the model perform if replacing all negative ReLU with positive ones and vice versa? What features are lost and how does it show up in the image space? (e.g. is it like residual artifacts? etc.)

The paper also mentions the design alleviates dead ReLU. Can the two branch use a Leaky ReLU? What would be the difference? There are many interesting questions raised from the proposed framework, but the paper seems to only touch on a few. I'd hope to get more insights from the authors and how they made their design choice of only experimenting with ReLU and how they are thinking about this problem.

**Others**

Line 102 mentions the iterative approach suffers from accumulated errors. However, how is the proposed two-stage method handling this problem, given the second stage also depends on the output from the first stage?


## **Post-rebuttal**

The authors have addressed my concerns with more quantitative results and intuitive explanations. I'd encourage the authors to add some of these explanations into the main paper for clarity. I support acceptance of this paper. The simple design of positive-negative ReLU to preserve all features in the network is interesting and novel, and can potentially inspire further research works.


**Time Spent Reviewing:**

2.5

---

> ### Author Response · Authors · 2021-08-10
> **Responses to the comments from reviewer myi5**
>
>
> First of all, thanks for your positive feedback on the novelty and potential of this work. We provide the responses to the main concerns from the reviewer as follows.
>
> **Q1**: Limitation of the model?
>
> **A1**: We respectfully argue that the results shown in the supplement, although being not that thoroughly separated, are not worse than the other methods, with reflections less left and textures of transmission better preserved. We sort all the testing samples by their PSNR metrics and report the numbers of samples smaller than 20.0 as below. We have analyzed the main source of the influence on the estimation by our YTMT-UCT. It can be seen that we have fewer bad samples compared with ERRNet on all the testsets though, several samples are still challenging to ours, 1) Strong reflections that violate the linear combination; and 2) Intensive overlaps of two layers, especially with those transmission layers having pure color backgrounds. For the former situation, a possible solution is to mask regions interfered by the strong reflection (upper truncated), and then adopt inpainting techniques to fill out the region. The latter problem can be somehow solved by introducing synthesized data with pure color backgrounds. But, in the submission, our main focus is on the effectiveness of the YTMT strategy, to avoid the influence of other factors and guarantee the fairness of comparison, we omit these attempts for further boosting the separation quality. We will add more words on the limitation of our method and provide some directions for future work in the next version.
>
>
>
> | Datasets       | Models   | Num. of bad samples (file names)  | Main problems                                                |
> | -------------- | -------- | -------------------- | ------------------------------------------------------------ |
> | Real20 (20)    | ERRNet   | 8 (15,87,103,...)    |                                                              |
> |                | YTMT-UCT | 5 (110,87,22,...)    | Residual reflections due to strong reflection and large overlap. |
> | Objects (200)  | ERRNet   | 10 (98,99,96,...)    |                                                              |
> |                | YTMT-UCT | 2 (98,99)            | Residual reflections due to large overlap of reflection and pure color background |
> | Postcard (199) | ERRNet   | 16 (176,85,177,...)  |                                                              |
> |                | YTMT-UCT | 11 (179,191,181,...) | Wrong removal of the sky region (some are corrected by the second stage). |
> | Wild (55)      | ERRNet   | 10 (49,30,26,...)    |                                                              |
> |                | YTMT-UCT | 9 (49,26,23,...)     | Residual reflections due to large overlap of reflection and pure color background. |
>
>
>
> **Q2**: What happens if the model only keeps the negative ReLU or only the positive ReLU?
>
> **A2**: The Concat. UNet is exactly the YTMT-UCS with all the negative ReLUs replaced by the normal ones. In this situation, the equation (2) will degenerate to $\mathbf{x}_1^l = \mathrm{ReLU}(\mathbf{\tilde{x}}_1^l)\oplus \mathrm{ReLU}(\mathbf{\tilde{x}}_2^l) = \mathbf{x}_2^l$. The two streams in the network will have the same activations, leading to degraded performance (see Table 3). Practically, we observe that YTMT-UCS produces mostly reasonable results, while Concat. UNet produces nearly all-zero reflection layers on many testing samples at the same training epoch. The residual reflections on the transmission layer will subsequently increase. Along with the training process, the residual reflections will be first reduced and then enlarged due to the nearly all-zero reflection layer and the last term of the equation (3). The all-positive and all-negative manners will have the same convergence behavior along with the SGD process.
>
> **Q3**: Can the two-branch use a Leaky ReLU? More insights?
>
> **A3**: Generally speaking, our strategy can be compatible with most, if not all, of the activation pairs (e.g. Softmax and Softmin). But for the additive problems, the pair of ReLU and negative ReLU is more suitable due to its "either A or B" nature that satisfies the task of SIRS. Without any efficient way to exchange the information, only using the Leaky ReLU is not enough to obtain favorable separations. Experimentally, we have developed a UNet with one encoder and two decoders (with a competitive amount of parameters), and its rectifiers are all changed into the Leaky ReLU. The results are shown below, and for the learning of the transmission layer, Leaky ReLU may bring more residual reflections because of the unfiltered leakage. Yes, as pointed out by the reviewer, there are many interesting aspects to discover and discuss, like the possibility of developing dual-ReLU blocks for high-level tasks (e.g. classification) or tasks suffering from sparse gradients, and other options for the activation. But, due to the limited space, we concentrated on the effectiveness of the YTMT strategy with simple designs. It is positive to envision how this designing concept can be similarly re-hashed into other kinds of tasks, or even be improved in terms of what embodies those networks (parts). As such, we think this work would bear significance in terms of its extensibility to many other tasks and applications in the future, and it would be valuable to the community as a whole.
>
> | Datasets       | Metrics | LeakyUNet2Tail | YTMT-UCS  |
> | -------------- | ------- | -------------- | --------- |
> | Real20 (20)    | PSNR    | 22.24          | **23.05** |
> |                | SSIM    | 0.791          | **0.805** |
> | Objects (200)  | PSNR    | 22.49          | **24.46** |
> |                | SSIM    | 0.874          | **0.891** |
> | Postcard (199) | PSNR    | 20.14          | **22.66** |
> |                | SSIM    | 0.859          | **0.885** |
> | Wild (55)      | PSNR    | 23.05          | **25.24** |
> |                | SSIM    | 0.867          | **0.887** |
>
> **Q4**: How is the proposed two-stage method handling the accumulated errors?
>
> **A4**: The latter stages of BDN and IBCLN have fewer chances to further exchange their miss-placed regions in the previous separations, given there is no explicit information interaction mechanism in the feature space of their network design. However, in the YTMT strategy, there are more chances for the network (e.g. 9 times in the YTMT-UCT). Moreover, the separations are estimated in a single network, rather than several sub-networks. We train the first stage of the network to converge, and then use it to initialize the second stage. The following process depends on not only the previous separations, but also the knowledge learned in the first stage. If there is no better separation the network can exploit, the second stage will produce a solution close to the outputs of the first stage. You can find the phenomenon that when iteratively feeding the output again into the trained network of Zhang et al. and ERRNet, the output transmission layer will converge to a stable result.
>
>
> We sincerely appreciate your suggestions, and will definitely proofread and revise the paper according to the comments, discuss and cite missing references,  to improve its quality.

---

### Official Review · Reviewer_zUwG · 2021-07-17

**Rating:** 6
**Confidence:** 5

**Summary:**

In this work, the authors address the problem of how to get a good approximation of a scene affected by reflections produced by a pane of glass. To do this, the authors proposed a dual-stream decomposition strategy called YTMT, where the main idea is how to effectively exchange information to enforce a better approximation of the target scene (and also of the reflection layer). At the optimisation level, the authors use a composite loss function which involves the reconstruction term along with three additional terms to enforce better output quality based for instance in constraining the gradients of the image.  Experimental setting shows readily competing results with respect to the existing techniques for reflection removal. Moreover, authors show the potential of the technique for other tasks such as blind denoising.



**Limitations And Societal Impact:**

There is not a clear discussion on the limitations of their work.

**Main Review:**

[Strengths]
--   The main strength of the paper is the idea of how to efficiently exchange/use information through a dual-stream decomposition strategy.  Moreover, the experimental results are competitive wrt the body of literature.

[Weaknesses]
------ A major point that turns down somehow the paper is the lack of formalism and intuition of the technique. More precisely:

-- The authors consider the image reflection model as a linear superposition between two layers I=T+R. However, different works, even early ones in the area, for  example [*] [**] showed that the reflection is less in focus and often blurred then considering in the model a weighting factor w regarding the strength of the reflections and a blurring kernel k. Why do the authors not consider this more realistic scenario as a model?

[*] Y.  Schechner,  N.  Kiryati,  and  R.  Basri,  “Separation  of  transparent layers  using  focus,”International  Journal  of  Computer  Vision  (IJCV),vol. 39, no. 1, pp. 25–39, 2000.
[**]Li  and  M.  S.  Brown,  “Single  image  layer  separation  using  relative smoothness,”  inIEEE  Conference  on  Computer  Vision  and  PatternRecognition (CVPR), 2014, pp. 2752–2759.

-- The main driving optimisation (7) is a combination of existing works and ideas that has been already applied to the task at hand including the gradients constraint and the exclusion term. Firstly, can the authors provide further thoughts on the level of novelty on the loss? Secondly, authors argue that they use an adversarial loss to ‘regulate the solution on the image manifold’  Can the authors argue where is coming that regularisation and where the space of images is modelling as a manifold?

-- The authors in the introduction motivate the approach and discuss an inherent ‘error/residual Q’. How does it fit into the big picture of the main driving optimisation and how Q is obtained?

-- From (3) authors set ‘a’ ‘b’ ‘c’ and ‘d’ empirically but what  was the range of values searched to set such values? Similarly for the set of  \lambda in (7). What is the intuition behind those parameters wrt the performance (e.g. what is the effect of increasing or decreasing them)?


------------ [Experimental Results] Although the experimental setting has merit, there are some points to clarify.
-- The authors are using SSIM and PSNR metrics for the quantitative results. However, several works including [a] and [b] have discussed that need for more perceptual meaningful metrics such as the sLMSE [b]. Why not using more perceptural metrics such as the sLMSE? That avoid for example  that a single edge error to dominate the overall performance

-- What is the motivation of Table 2? One is interested in recovering the target scene rather than the reflection itself. Whilst the reflection layer indeed helps in the model formulation the relevant output is the target scene (transmission layer).

-- From Table I, It is unclear the level of performance gain from the proposed technique. The numerics seem not to have a statistical significant difference wrt ERRNet (and Lei et al). Authors should argue the main benefit of the proposed technique.

-- The authors in the opening of the paper (footnote 1 in page 1) mentioned that the proposed technique can be applied to other tasks but they are not being explored in the paper as the main topic is reflection removal. However, authors actually include experiments regarding blind denoising -- Therefore, the title is not matching the content of the paper. Authors need to further motivate the inclusion of section 5


[a]Wan, R., Shi, B., Duan, L. Y., Tan, A. H., & Kot, A. C. (2017). Benchmarking single-image reflection removal algorithms. In Proceedings of the IEEE International Conference on Computer Vision (pp. 3922-3930).
[b] Grosse, M. K. Johnson, E. H. Adelson, and W. T. Free-man. Ground truth dataset and baseline evaluations for in-trinsic image algorithms. InProc. ICCV, 2009.



------------------------------------------------------------
POST-REBUTTAL

After reading the rebuttal carefully, the authors addressed my concerns and even provided further experiments to further support their statements. * And additional minor comment is regarding Q9, what I wanted to covey was that, from a reader point of view, it would be better to have only results on the task of reflection removal rather than adding another task (blind denoising) with a short description. Overall, I’m more positive about this paper than before and this is therefore reflected in the final suggested score -- expecting the authors to implement the points from the rebuttal in a the revised version for completeness.





**Time Spent Reviewing:**

4

---

> ### Author Response · Authors · 2021-08-10
> **Responses to the comments from reviewer zUwG**
>
> First of all, thanks for your effort on reviewing our paper. We provide the responses to the main concerns from the reviewer as follows.
>
> **Q1**: Why not use the reflection model as $I = T + k*R$?
>
> **A1**: As pointed out by the reviewer, yes, some works formulate the model as $I = T + k * R$. Without involving the deblurring operation, the mentioned manner is nothing different from $I=T+R$ in practice. In other words, by focusing on the separation task, we can treat $k*\tilde{R}$ as $R$, which acts as one component from mixed signals. In fact, the adopted datasets contain synthetic data generated by blurring the reflection using different kernels. Our YTMT can accomplish the job as verified in the experiments. Therefore, we use $R$ here to express the reflection layer as many other methods do [a, b].
>
> [a] Zhang, Xuaner, Ren Ng, and Qifeng Chen. "Single image reflection separation with perceptual losses." CVPR. 2018.
>
> [b] Wei, Kaixuan, et al. "Single image reflection removal exploiting misaligned training data and network enhancements." CVPR. 2019.
>
> **Q2**: Novelty of the loss?
>
> **A2**: To validate our main proposal/contribution, an interactive dual-stream interactive **strategy design**, and largely exclude the influence from the losses, we choose the similar loss setting with ERRNet. Our overall superiority over ERRNet clearly reflects the effectiveness of our strategy. The novelty or contribution of one work may not always come from the loss design, for instance ResNet, attention module, normalization manner, and many others.
>
> **Q3**: Regularization on the manifold?
>
> **A3**: As previous work discussed [a,b], GAN is a good approximator for natural image manifold. In our work, the manifold of ground-truth transmission layer is learned by the generative adversarial framework. As a manifold reference, the restoration of the transmission layer is expected to be around the manifold of the ground-truth transmission layer. Moreover, the benefits of using a GAN framework in the SIRS task have been well demonstrated in prior methods like [c,d].
>
> [a] Pan, Xingang, et al. "Exploiting deep generative prior for versatile image restoration and manipulation." ECCV, 2020.
>
> [b] Menon, Sachit, et al. "Pulse: Self-supervised photo upsampling via latent space exploration of generative models." CVPR, 2020.
>
> [c] Zhang, Xuaner, Ren Ng, and Qifeng Chen. "Single image reflection separation with perceptual losses." CVPR. 2018.
>
> [d] Wei, Kaixuan, et al. "Single image reflection removal exploiting misaligned training data and network enhancements." CVPR. 2019.
>
> **Q4**: How the "trash" $Q$ is obtained and fitted into the main driving optimization?
>
> **A4**: $Q$ is the part of deactivated features by the ReLU function and re-collected by the negative ReLU, and then transferred into the other branch. This is exactly what we do (the YTMT) in this work. We have illustrated this point **throughout** the whole paper. We here emphasize again that, the $Q$ is obtained by the negative ReLU, and fits into the main optimization by interaction.
>
> **Q5**: The intuition behind the hyper parameters?
>
> **A5**: Assuming the total amount/intensity of the pixels of $R$ and $\nabla T$ are often weaker than $T$, we need to give them larger weights to balance our attention to the parts with content. The item $||\hat{T}+\hat{R} - I||_{1}$ works as a constraint to the decomposition, its weight should be a little smaller than the weights of $\hat{T}$ and $\hat{R}$. As for the $\lambda$, the value range of perceptual loss and adversarial loss is much larger than the pixel loss, so we need to give them smaller weights to balance them and the ERRNet has a similar combination. We note it again that the design of the loss function is not our main concern, and it is possible to find a more suitable hyper parameter combination by searching from a large parameter space.
>
> **Q6**: Can you provide more perceptual metrics?
>
> **A6**: As suggested by the reviewer, we have evaluated the results in terms of NCC and sLSME. Our sLMSE metrics are 0.977, 0.997, 0.996 and 0.992 for real20, objects, postcard and wild respectively, and NCC metrics are 0.895, 0.982, 0.945 and 0.927, respectively. Except for minor differences, they are basically subject to the order of PSNR and SSIM. Notably, our NCC metric is higher than ERRNet and BDN for 2\% and 10\% on real20, respectively, and 19\% higher than BDN on objects, with the sLMSE metrics staying the best. We will add the results in terms of these metrics in the future version.
>
> **Q7**: Necessity of Table 2?
>
> **A7**: Generally speaking, each component for blind source separation tasks should be treated equally. For example, the intrinsic image decomposition task decomposes an image into reflectance and shading, both of which are important. As for the SIRS task, some individuals like the reviewer may pay more attention on the transmission layer for some specific demands, while someone else may highlight the usage of the reflection layer. In other words, the reflection layer can be a critical part of the information to reproduce the image shooting scene, as discussed in [a] (also in the 1st paragraph of Introduction of the submission).  In Section 2.3 of [b], the occluding foregrounds, including reflections, are super-imposed back to the edited regions to achieve a more realistic visual effect. The favorable reflection layers are also desired in [c]. Moreover, from the perspective of optimization, a closer solution of one variable to the optimum is definitely beneficial to the whole procedure. Despite different demands from different users, it is nothing harmful to recover two layers with high quality in comparison with one of them. This is the main motivation of Table 2.
>
> [a] Wan, Renjie, et al. "Reflection scene separation from a single image." CVPR, 2020.
>
> [b] Guo, Xiaojie, et al. "Video editing with temporal, spatial and appearance consistency." CVPR, 2013.
>
> [c] Xue, Tianfan, et al. "A computational approach for obstruction-free photography." ACM TOG 34.4 (2015): 1-11.
>
> **Q8**: Difference between your method and the others in Table 1?
>
> **A8**: As all the other reviewers noticed, our superiority on SIRS over other state-of-the-art methods is clear and comprehensive, both quantitatively and qualitatively. One can also refer to Figure 4 and Figure 5 as examples to find out the obvious visual advantages of our method on different testsets at the same time. A main challenging factor of the current SIRS task is to obtain higher metrics on both real20 and the three subsets of the $SIR^2$ dataset at the same time, given all these datasets have completely different data distributions. IBCLN has only one metric on one subset being higher than ours, but with a large margin with ours on the others, especially on the real20 testset, and the ERRNet produces unsatisfactory results on the wild subset. The margin between Lei et al. and ours on all testsets is large enough to clearly illustrate our superiority. More comparisons made on real45 also support our better performance, which can be found in the supplementary material.  Moreover, the restoration quality of reflection layers of the ERRNet and IBCLN is far from satisfactory compared with our method, the importance of which has been already highlighted in **A7**.
>
> **Q9**: Where are the applications of other tasks you mentioned on page 1?
>
> **A9**: Due to the page limit, we have put some applications in the supplementary material. You can check it if interested. We are sorry for being impossible to elaborate all possible applications.
>
> We sincerely appreciate your suggestions, and will definitely proofread and revise the paper according to the comments to improve its quality.

---

### Official Review · Reviewer_XY18 · 2021-07-18

**Rating:** 6
**Confidence:** 4

**Summary:**

This paper proposes a dual-stream network design equipped with an interactive strategy (coined YTMT) to tackle the blind source separation tasks, especially the single image reflection separation. The underlying design motivation comes from a very simple mathematical derivation, i.e., any information discarded from one layer might be beneficial to reconstruct another layer. They demonstrate performance improvement on the task of single image reflection removal. The possible extensions to other source separation tasks, e.g., the blind image denoising and intrinsic image decomposition, are also investigated.

**Main Review:**

## Pros
* The introduction part is well written; the design motivation is clear and reasonable.
* Good performance on SIRS against state-of-the-art methods.
* The potential applications to other blind source separation tasks.

## Cons
Though the underlying principle looks sound, it largely relies on the assumption that blind source separation could be better handled by  dual-stream networks. Such an assumption is, however, not established in majority of source separation tasks. To separate two layers, one may argue only a single network with two output branches at tail is sufficient to interact information within shared backbone in an implicit manner. The authors should consider adding such an ablation study (with comparable computation overhead) to demonstrate the benefit of dual-stream network design.

Besides, in Table 3, the performance of two-UNet should be identical to a single UNet, as there is no interaction from another stream, thus the comparison there is not so meaningful. The descriptions of Split UNet and Concat. UNet are also unclear, the authors should give some explanations on why they design such architectures for ablation. The ablation study in Table 4 looks redundant, since neither the two stage design nor the attention module is the authors' contribution. The authors should reorganize the paper by moving other content, e.g., the intrinsic image decomposition, from the suppl. material into the main paper.

## Summary
Overall, the proposed idea is simple yet reasonable, but more experimental verifications are needed to justify its superiority.
I tentatively give a borderline score and would change it according to the authors' response.

## Post rebuttal
After reading the rebuttal and other reviewers' comments, I think most issues raised by the reviewers are repairable, and many of them have been already addressed in the rebuttal. As a result, I lean to borderline accept and it's ok for me if the rejection is made on this paper.
Also, I highly recommend the authors refine the method and experiment parts, particularly, describing the abbreviations (e.g., Concat. UNet, YTMT-UCS) carefully to avoid potential confusion and distraction.

**Time Spent Reviewing:**

3

---

> ### Author Response · Authors · 2021-08-10
> **Responses to the comments from reviewer XY18**
>
> First of all, thanks for your positive feedback on the novelty and potential of this work. We provide responses to the main concerns from the reviewer as follows.
>
> **Q1**: Is a single network with two output branches at tail  (1Net2Tail for short)  sufficient to interact information?
>
> **A1**: The effectiveness of 1Net2Tail is inferior to YTMT. The main reason is that the non-guaranteed information separation ability of two layers, which is the key motivation of our YTMT design. 1) If merely using the convolution mechanism, the information can be hardly separated in the main branch; 2) While introducing the activation like ReLU into the simple convolution manner in the main branch, the disactivated information will be simply discarded; 3) For the quality of reconstruction of the target layers, the feature channels inevitably need to be expanded to avoid information loss, because there is no information interaction between the two tail branches. Otherwise, the information would be insufficient to restore the two components. Our YTMT is a strategy designed for mitigating the above issues in an effective and efficient way. Meanwhile, the reflection branch can easily produce zero outputs due to the weak ground-truth reflection (intensity) and the lack of auxiliary information from the other branch. As mentioned by the reviewer, yes, it seems reasonable for two tails to select and produce their own information from a common feature set (main branch), but the above 3 issues remain. To make the claim more convincing, as suggested by the reviewer, we conduct the following experiment. A UNet with one encoder and two decoders is built with a similar amount of parameters as the YTMT-UCS. The numerical comparison is given in the following table, from which, we can see the clear advantage of the YTMT over the 1Net2Tail.
>
>
>
> | Datasets       | Metrics | 1Net2Tail | YTMT-UCS  |
> | -------------- | ------- | --------- | --------- |
> | Real20 (20)    | PSNR    | 22.16     | **23.05** |
> |                | SSIM    | 0.799     | **0.805** |
> | Objects (200)  | PSNR    | 23.85     | **24.46** |
> |                | SSIM    | 0.881     | **0.891** |
> | Postcard (199) | PSNR    | 21.13     | **22.66** |
> |                | SSIM    | 0.860     | **0.885** |
> | Wild (55)      | PSNR    | 24.57     | **25.24** |
> |                | SSIM    | 0.877     | **0.887** |
>
>
>
> **Q2**: Is the two-UNet identical to a single UNet?
>
> **A2**: No, they are actually not identical. Let us consider only estimating one component, say $\hat{T}$ (the other is similar). The single UNet would only take the supervision from the ground-truth $T$. And the reflection is obtained by $I-\hat{T}$, the loss $||\hat{T}+\hat{R} - I||$ will be always 0. This does not matter if the estimation $\hat{T}$ is “perfectly” $T$, which however is barely the case. If taking the two-UNet, the estimations of the two layers are highly related to each other. The separation solutions are therefore supervised by the ground-truth of both transmission and reflection layers. Further, the optimization constrained by multiple regularizers are more likely to improve the generalization ability of the network. The improvement can be observed in Table 5, which says that DnCNN-2 achieves better performance against the DnCNN when the noise levels are out of the training scope.
>
> **Q3**: Why design Split UNet and Concat. UNet for the ablation?
>
> **A3**: The Concat. UNet is actually YTMT-UCS with all the negative ReLUs replaced by the normal ReLUs to validate if the YTMT strategy really works to preserve the discarded information. However, by doing this, the different branches actually have  same features, which is clearly not beneficial to the layer decomposition. Thus we develop another baseline architecture named Split UNet, which transfers half of the feature channels, rather than all the channels like Concat. UNet (e.g. transferring the latter 32 channels of the feature map to the other branch, and preserving the former 32 channels, given 64 channels activated by the normal ReLU). In other words, this design exchanges half of the ReLU activation between two branches, avoiding the completely same features in the different streams. However, we can see from Table 3 that it still falls behind the YTMT design, and even the Concat. UNet. This may result from the smaller number of channels in the normal stream. In summary, our YTMT strategy is a more reasonable and efficient solution compared with the baselines. The designs of the interaction blocks of Split UNet and Concat. UNet can be found in Figure 1 of the supplementary material.
>
> **Q4**: Is the ablation study in Table 4 redundant?
>
> **A4**: Please notice that the notation “A” in YTMT-UAS and YTMT-UAT is the additive YTMT block as shown in Figure 1 (c), instead of the attention module. We respectfully argue that the ablation study in Table 4 is necessary. Since both the concatenated and additive modules fit our YTMT strategy, as two possible options, we compare their performance in Table 4 for giving a clue to readers. Moreover, in case of the concerns like “To what extent does the two-stage scheme improve the performance of your model” and “How about your model performs with a single stage”, we show the comparison of using two-stage or not in Table 4.
>
> We sincerely appreciate your suggestions, and will definitely proofread and revise the paper according to the comments to improve its quality.

---

### Decision · Program_Chairs · 2021-09-27

**Decision:**

Accept (Poster)

**Comment:**

The reviewers in general liked the idea and results of the paper: simple designs of keeping all features in the network by adding dual-ReLU branch; good results on SIRS against SOTA. Many issues were raised, e.g. lack of comparison, formalism and intuition of the technique. The reviewers were satisfied with the rebuttal and bumped up the score. Please incorporate the post-rebuttal comments from the reviewers, e.g. having only results on the task of reflection removal rather than adding another task with a short description (can use the space to make the figures larger), and improving the text and simplifying the figures.